# Leptin in Atherosclerosis: Focus on Macrophages, Endothelial and Smooth Muscle Cells

**DOI:** 10.3390/ijms22115446

**Published:** 2021-05-21

**Authors:** Priya Raman, Saugat Khanal

**Affiliations:** 1Integrative Medical Sciences, Northeast Ohio Medical University, Rootstown, OH 44272, USA; skhanal@neomed.edu; 2School of Biomedical Sciences, Kent State University, Kent, OH 44240, USA

**Keywords:** hyperleptinemia, endothelial cells, vascular smooth muscle cells, macrophages, atherosclerosis

## Abstract

Increasing adipose tissue mass in obesity directly correlates with elevated circulating leptin levels. Leptin is an adipokine known to play a role in numerous biological processes including regulation of energy homeostasis, inflammation, vascular function and angiogenesis. While physiological concentrations of leptin may exhibit multiple beneficial effects, chronically elevated pathophysiological levels or hyperleptinemia, characteristic of obesity and diabetes, is a major risk factor for development of atherosclerosis. Hyperleptinemia results in a state of selective leptin resistance such that while beneficial metabolic effects of leptin are dampened, deleterious vascular effects of leptin are conserved attributing to vascular dysfunction. Leptin exerts potent proatherogenic effects on multiple vascular cell types including macrophages, endothelial cells and smooth muscle cells; these effects are mediated via an interaction of leptin with the long form of leptin receptor, abundantly expressed in atherosclerotic plaques. This review provides a summary of recent in vivo and in vitro studies that highlight a role of leptin in the pathogenesis of atherosclerotic complications associated with obesity and diabetes.

## 1. Introduction

Cardiovascular disease (CVD) is the leading cause of death world-wide accounting for 18.6 million deaths in 2019, a number expected to increase to greater than 23.6 million by 2030 [1]. Atherosclerosis is a major player in the development of several cardiovascular complications including myocardial infarction, angina and heart failure. Over the past two decades, the global incidence of obesity and Type 2 diabetes has escalated at alarming rates. Multiple lines of evidence support the notion that obesity and Type 2 diabetes are major risk factors for development of atherosclerosis [2,3,4,5,6,7,8]. Obesity, itself, is a well-known trigger for Type 2 diabetes [9]. Vascular disease is the most rampant and deadliest of all health concerns affecting diabetic and obese populations in our society. Risks of atherosclerotic complications are augmented two-to-four-fold in diabetic and obese patients [10,11], accounting for increased morbidity and mortality among those individuals. However, the fundamental mechanisms that drive accelerated vasculopathy in obesity and diabetes remain incompletely understood. The aim of this review is to comprehensively discuss the role of an adipose tissue-derived hormone, leptin, in the etiology of vascular dysfunction.

## 2. Obesity, Adipose Tissue and Cardiovascular Diseases

Obesity results from an imbalance between caloric intake and energy expenditure, and directly correlates with increasing adipose tissue mass characterized by adipocyte hypertrophy and adipocyte hyperplasia [12]. Adipose tissue dysfunction is a key determinant of obesity-associated metabolic abnormalities [12,13]. Although previously viewed as an inert storage depot devoid of any metabolic activity, the emergent role of the adipose tissue as an organ with endocrine function is well established. A plethora of studies have revealed that the adipose tissue, particularly white adipocytes, secrete numerous proteins collectively known as the ‘adipokines’ with varied endocrine functions [14]. Such adipocyte-secreted proteins play an important role in the regulation of glucose and fatty acid metabolism, energy homeostasis, inflammatory responses as well as control of blood pressure [14,15,16,17]. A few classic examples of adipokines are the pro-inflammatory proteins such as leptin, resistin, tumor necrosis factor-alpha (TNF-α) and interleukin-6 (IL-6) that may result in adverse metabolic and immune responses, as well as anti-inflammatory proteins such as adiponectin. There is growing evidence linking increasing visceral adiposity with differential secretion of adipokines [18,19]. An imbalance in the circulating concentrations of one or more adipokines may contribute to the pro-inflammatory state characteristic of obesity, in turn providing a basis for obesity-related complications [20,21]. While the current literature has many excellent reviews discussing the contribution of different adipokines in the etiology of obesity and related metabolic and cardiovascular disorders, the role of leptin in pathogenesis of atherosclerosis has been somewhat conflicting. The current review attempts to provide an overview of recent in vivo and in vitro studies highlighting the involvement of leptin in pathogenesis of atherosclerotic complications associated with obesity and diabetes.

## 3. Leptin—Friend or Foe in Atherosclerosis and Vascular Dysfunction

### 3.1. Clinical Studies

Leptin, initially discovered as a satiety factor regulating food intake and energy expenditure [22], is a pleiotropic adipocyte-secreted hormone that participates in multiple biological processes including inflammatory response, immune function, angiogenesis, vascular function, bone homeostasis and reproduction [23]. Elevated serum leptin levels directly correlate with increasing adipose tissue mass and is a major driving force for obesity and related metabolic perturbations. Epidemiological evidence indicates an association between hyperleptinemia and several cardiovascular diseases including congestive heart failure, myocardial infarction, hypertension and coronary artery disease [24,25]. These findings suggest an imperative role of leptin linking obesity and related metabolic disorders with vascular dysfunction. Multiple clinical and experimental studies (reviewed in [26,27]) underscore hyperleptinemia, a hallmark of obesity, as an independent risk factor for development of macrovascular complications [26,27,28]. For instance, in a cross-sectional study of Type 2 diabetes, plasma leptin levels were associated with coronary artery calcification (CAC), a measure of coronary atherosclerosis, after adjusting for adiposity and C-reactive protein (CRP) [28]. A study conducted by McMahon et al. [29] demonstrated that high leptin levels increased risk of subclinical atherosclerosis in systemic lupus erythematosus, and this was accompanied with elevated levels of inflammatory biomarkers such as Lp(a) and oxPL/apoB100. In a later study, increased waist circumference and augmented serum leptin levels occurred concomitant to elevated serum cholesterol, triglyceride and CRP expression in coronary artery disease (CAD) patients compared with healthy controls [30]. A strong, positive correlation was also observed between elevated plasma leptin levels and the number of stenotic coronary vessels in CAD patients [31]. More recently, it was reported that augmented serum leptin confers a much greater risk for development of calcific aortic vascular disease; this detrimental effect of leptin was found to increase as a function of age and reduced renal function [32]. These data are consistent with earlier reports wherein serum leptin levels in obese or overweight individuals were shown to be predictive of myocardial infarction, atherosclerosis, stroke or coronary events [30,33,34]. Contrary to these findings, few other studies have revealed vasculoprotective effects of leptin in obese humans. Specifically, a meta-analysis including 13 epidemiological studies demonstrated an inverse relationship between leptin levels and coronary heart disease (CHD), and these data were consistent across both male and female patients [35]. The studies included in this meta-analysis were highly comparable in terms of their cohort composition based on either CHD or stroke as the type of outcome. Moreover, all included studies were adjusted for sociodemographic (age, race, town) as well as other common cardiovascular risk factors (diabetes, lipids, BMI, hypertension, smoking status). These results led to the idea that elevated leptin levels may not be associated with risks of developing CHD or stroke. While these findings were consistent with a few earlier reports showing lack of any significant association between leptin levels and risk of CVD events, other studies have implicated leptin as an important risk factor for CVD. However, as noted by the authors, this meta-analysis was reported to have a few heterogeneities in terms of sample size, number of events and differences in leptin levels compared to values that typically evolve in obesity; accordingly, such differences possibly attribute to the observed discrepancies in results. In a more recent cross-sectional study conducted in young normotensive individuals, circulating leptin levels showed a beneficial inverse relationship with the carotid intima-media thickness and cross-sectional wall area (surrogate markers of carotid wall thickness); interestingly, in this study an association between leptin levels and measures of subclinical atherosclerosis was noted only in overweight individuals [36]. Taken together, these clinical data (summarized in Table 1) point out considerable incongruity about the role of leptin in vasculopathy.

### 3.2. Animal Studies

In line with the appreciable amount of clinical data illustrating hyperleptinemia as a link between obesity and vascular dysfunction, animal studies using vascular injury models have shown that exogenous leptin promotes neointimal growth and vascular remodeling [37,38,39]. Consistently, vascular injury mouse models of obesity and diabetes have demonstrated that leptin deficiency or leptin receptor signaling defects protect against neointimal hyperplasia [40,41]. On the other hand, leptin-induced neointimal formation was shown to be completely blunted in ApoE^-/-^ mice suggesting that leptin requires ApoE to promote vascular lesions [42]. Rodent models of diabetes and obesity have yielded inconsistent data about leptin’s involvement with atherosclerosis. An earlier study by Bodary et al. [43] reported that administration of recombinant leptin increased the atherosclerotic lesion area in carotid and brachiocephalic arteries of ApoE^-/-^ mice, in the absence of significant differences between total cholesterol and total triglyceride levels. In accordance with these findings, Chiba et al. [44] reported that hyperlipidemic leptin-deficient ob/ob;ApoE^-/-^ mice developed reduced atherosclerotic lesions primarily comprising of fatty streaks, despite elevated triglyceride levels, compared with ApoE^-/-^ mice with intact leptin; moreover, these changes occurred independent of serum cholesterol and pro-inflammatory markers. Similar observations were also noted in leptin-deficient ob/ob and leptin receptor-deficient db/db mouse models in the background of LDL-R knockouts [45]. Recent data from our laboratory lend strong support to these lesion-promoting properties of leptin. Specifically, we have shown that exogenous leptin at pathophysiological concentrations leads to robust lipid-filled lesions with increased collagen content accompanied with elevated total triglyceride levels in ApoE^-/-^ mice, despite lack of significant changes in plasma total cholesterol [46]. Together, these results suggest a regulatory role of leptin in pathogenesis of atherosclerosis. Contrary to these findings, Wu et al. [47] previously demonstrated profound increase in plasma total cholesterol and triglyceride levels accompanied with enhanced aortic lipid content in db/db;ApoE^-/-^ vs. age-matched ApoE^-/-^ littermates; these data have suggested that absence of functional leptin receptor accelerates atherosclerosis. Congruently, a dose-dependent reduction in atherosclerotic plaque area was noted in ob/ob;LDL-R^-/-^ mice following leptin administration within supraphysiological to physiological levels. This protective outcome of leptin was shown to be mediated via reduced total cholesterol and LDL-cholesterol levels, downregulation of hepatic genes involved in lipid metabolism and inflammation as well as upregulation of adiponectin [48]. In accordance with these vasculoprotective effects of leptin, Jun et al. [49] reported that administration of low-dose leptin, mimicking physiological conditions, remarkably attenuated LDL-cholesterol and atherosclerotic lesions in Ins2^+/Akita^/ApoE^-/-^ mice, a model of spontaneous Type 1 diabetes that develops hypercholesterolemia and atherogenesis. Moreover, this effect was shown to be mediated via restoration of hepatic insulin receptor signaling. Collectively, these results have indicated a therapeutic value of physiological leptin on the cardiovascular burden associated with Type 1 diabetes, possibly mediated via partial or complete restoration of altered gene expression patterns of fatty acid metabolism and lipogenesis. Such disparate findings in the current literature (summarized in Table 2) have largely confounded our understanding of the role of leptin in atherosclerosis.

The observed discrepancy in the beneficial vs. detrimental effects of leptin on development and progression of atherosclerosis may be due to differences in leptin concentration (physiological vs. supra-physiological) and animal models (lean vs. obese phenotype) utilized in various studies. Additional factors include differences between atherosclerotic mouse models (ApoE^-/-^ vs. LDL-R^-/-^), lesion stage (early vs. advanced), specific lesion site and confounding co-morbidities of the animal model studied (e.g., altered insulin signaling, proinflammatory milieu, etc.). As such, pathophysiological leptin levels appear to be the primary trigger for augmented vasculopathy associated with diabetes and obesity. Therefore, it would be reasonable to posit that while leptin may elicit a therapeutic response at physiologically relevant concentrations, this beneficial effect of leptin is completely abrogated in diseased states such as obesity and diabetes, characterized by hyperleptinemia. Under such conditions, elevated circulating leptin levels, likely arising as a compensatory mechanism to overcome reduced tissue sensitivity to leptin, represent a key driving force for development of atherosclerotic complications, independent of metabolic or inflammatory dysregulation.

## 4. Leptin Signaling—Metabolic vs. Vascular Dysfunction

It is well-established that leptin produces its biological response via interaction with the long form of the leptin receptor (Ob-Rb) triggering activation of multiple downstream signaling cascades including janus-associated kinase/signal transducer and activator of transcription (JAK/STAT), mitogen-activated protein kinase (MAPK) and insulin receptor substrate/phosphoinositide 3-kinase/Akt (IRS/PI3K/Akt), mTOR and AMP-activated protein kinase (AMPK) pathways [22,50]. Both human and rodent studies have documented an increased expression of leptin receptors in the injured vascular wall and atherosclerotic plaques [51,52,53]. Moreover, an interaction of leptin with its receptor was identified as the first critical step in atheroma formation [52]. Leptin has been shown to modulate the expression of several vascular genes associated with atherosclerosis and abnormal angiogenesis including cytokines, chemokines, growth factors and extracellular matrix proteins [54,55,56,57,58,59,60]. We previously reported that leptin, at concentrations relevant to obesity and diabetes, has a direct stimulatory effect on the proatherogenic matricellular protein, thrombospondin-1 (TSP-1), implicated in atherosclerosis. We have further shown that leptin-induced upregulation of TSP-1 requires the presence of intact leptin signaling specific to JAK2/ERK/JNK-dependent pathways [61]. These results were confirmed by recent in vivo findings wherein exogenous leptin, at concentrations mimicking obesity, accelerated atherosclerotic lesion formation concomitant to augmented TSP-1 expression in the vascular walls of ApoE^-/-^ mice with intact leptin receptors [46]. Cogent to previous studies, our results support a crucial role of the leptin signal transduction pathway in the development of atherogenesis. Contradictory to these data, earlier studies have reported increased aortic total cholesterol, free cholesterol and cholesteryl ester content in leptin receptor-deficient db/db;ApoE^-/-^ mice compared with ApoE^-/-^ littermates with intact leptin signaling [47]. While these observations may imply a conflicting role of leptin and leptin signaling in the pathogenesis of atherosclerosis, it is conceivable that obese and diabetic patients develop a state of ‘selective’ leptin resistance. Accordingly, while centrally mediated actions of leptin may be compromised in hyperleptinemic individuals, the peripheral effects of leptin are well-preserved. Along these lines, the hypothalamus, liver, pancreas and skeletal muscles are considered as leptin-resistant tissues mediating the metabolic effects of leptin. On the other hand, the vessel wall, myocardium, immune cells and blood platelets represent the non-leptin resistant tissues [62] that mediate the vascular functions of leptin. This leads to the concept that central leptin resistance may account for lack of hypothalamic regulation of food intake and energy expenditure as well as metabolic dysfunction, attributing to obesity, diabetes and other metabolic defects in hyperleptinemia. On the other hand, preservation of leptin signaling mechanisms in the vasculature may be responsible for hyperleptinemia-induced vascular dysfunction associated with obesity and diabetes (Figure 1). Thus, the existing dogma in the field is that while physiological leptin concentrations may exert endothelium-dependent vasodilatory effects, pathophysiological elevated leptin levels akin to obesity and diabetes exhibit multiple proatherogenic properties. Previous studies have reported that augmented leptin levels reduce nitric oxide availability and enhance superoxide generation attributing to impaired arterial vasodilation and reduced vessel distensibility in obesity [63,64,65]. Leptin is also known to have pro-coagulant [66,67], anti-fibrinolytic [68,69], pro-inflammatory [70] and pro-angiogenic [71] properties. Chronically elevated leptin concentrations increase oxidative stress, promote thrombus formation as well as facilitate vascular inflammation and atherosclerosis [64,72].

## 5. Basic Overview of Atherosclerosis

Atherosclerosis is a chronic inflammatory disorder of the arterial wall triggered by an initial vascular response to endothelial injury [73]. In a healthy blood vessel, the endothelial cell lining provides an active interface between the circulating blood and the vessel wall. The primary function of this endothelial monolayer is to regulate adequate fluid, nutrient and gas exchange between the blood and tissues. In addition, the endothelial lining of the vessel wall regulates multiple functions that include control of vascular tone, generation of potent vasoconstrictor (endothelin-1) and vasodilator compounds (nitric oxide, prostacyclin), coagulation, lipoprotein permeability as well as adhesion and migration of formed blood elements. Endothelial dysfunction is an early marker of atherosclerosis [74,75]. In response to proatherogenic stimuli (e.g., hyperglycemia, hyperlipidemia, obesity, to name a few), damage of the endothelial cell barrier triggers an increased expression of multiple adhesion molecules (VCAM-1, ICAM-1) on the endothelial cell surface. This in turn facilitates attachment of circulating monocytes to the damaged endothelium followed by transmigration of monocytes into the intimal layer underlying the subendothelial matrix of the vasculature. Additionally, changes in the shape and properties of the endothelial lining, such as loss of endothelial tight junctions, provide a gateway for circulating lipoprotein particles (LDL, VLDL), typically elevated in obesity and related metabolic conditions. Within the intimal layer of the vessel wall, oxidation of LDL lipids renders them atherogenic; infiltrated monocytes further differentiate into macrophages. This is followed by an uptake of the oxidized LDL by macrophages mediated via scavenger receptors (CD36, LOX-1, SR-A) occurring on the surface of macrophages. Formation of such lipid-laden macrophages or ‘foam cells’ represents the earliest stage of atherosclerosis, referred to as fatty streak formation [76]. Progression of atherosclerotic lesions involves release of several growth factors and cytokines by endothelial cells, T-lymphocytes and macrophages. These growth factors and cytokines mediate smooth muscle cell (SMC) migration from the medial layer of the vascular wall into the intimal layer followed by extensive SMC proliferation; this in turn profoundly increases SMC content within the subendothelial space, characterizing complex atherosclerotic lesions. Accumulating data indicate that such hyperproliferative SMCs synthesize different extracellular matrix macromolecules such as collagen, elastin and glycoproteins; in addition, SMCs can engulf oxLDL via scavenger receptors (LRP, CD36) located on their cell surface, triggering formation of SMC-derived foam cells in the subendothelial matrix [77]. This progressive increase in SMC abundance and lipid accumulation is a key feature of advanced atherosclerotic lesions that ultimately contributes to vascular occlusion and significant reduction in the arterial lumen.

## 6. Leptin and Macrophages

Macrophages are the major immune cell populations found in atherosclerotic lesions. These cells play an important role in lesion initiation as well as progression to more advanced plaques. Increased accumulation of macrophage-derived foam cells in the intimal layer of the arterial wall is a hallmark of atherosclerosis [78,79]. Cholesterol homeostasis is a key determinant of macrophage inflammatory status. It is suggested that reduced cellular cholesterol levels may contribute to macrophage polarization to an anti-inflammatory (M2) phenotype [80,81]. Leptin is recognized as a potent monocyte/macrophage chemoattractant [82]. Previous studies have shown that murine macrophages isolated from leptin-deficient ob/ob mice exhibit reduced cholesterol accumulation and lower inflammatory response, revealed via attenuated CD36, MHC Class II, CD11b, CD40 and SR-A expression [83]. These results have suggested that leptin deficiency decreases foam cell formation and inhibits development of atherosclerosis. Incubation with leptin at concentrations typically seen in obese patients was found to increase TNF-α, IL-6 and IL-1β protein expression in human monocyte-enriched mononuclear cells [84]. In a subsequent study utilizing primary cultures of human monocytes/macrophages [85], it was shown that physiological concentrations of leptin increased accumulation of cholesteryl ester via upregulation of acyl CoA:cholesterol acyltransferase-1 (ACAT-1), an enzyme catalyzing cholesteryl ester synthesis, in the absence of any effect on the endocytic uptake of acetyl-LDL. Leptin was also found to suppress HDL-mediated cholesterol efflux under these conditions. ACAT-1 also plays a role in differentiation of monocytes to macrophages. To this end, leptin-induced upregulation of ACAT-1 expression is postulated to facilitate monocyte-to-macrophage differentiation, prompting accelerated atherosclerosis. Growing literature indicates that the transcription factor PPAR-γ, a major target of thiazolidinedione family of anti-diabetic compounds, is a key regulator of macrophage lipid metabolism [86]. PPAR-γ is also involved in macrophage differentiation and inflammatory response. Previous work has demonstrated that in primary human macrophage cultures and macrophage-derived foam cells, elevated leptin levels reduce PPAR-γ mRNA levels [87]. These findings suggest leptin-mediated downregulation of PPAR-γ as a plausible mechanism underlying hyperleptinemia-induced vascular dysfunction. Taken together, these data provide evidence for leptin’s ability to promote proatherogenic macrophage formation. Few possible mechanisms include increased secretion of inflammatory cytokines, augmented proliferative capacity, oxidative stress and impaired lipid metabolism (Figure 2). As opposed to these in vitro results, an in vivo animal study by Surmi et al. [88] failed to support a role of macrophages in hyperleptinemia-induced atherogenesis. Briefly, hyperleptinemic obese LepR^db/db^;LDL-R^-/-^ mice, lacking functional leptin receptors, were subjected to bone marrow transplantation from donor mice with either intact leptin receptors (LepR^wt^) or dysfunctional leptin receptors (LepR^db/db^). Interestingly, no differences in aortic root lesion area could be detected between LepR^db/db^;LDL-R^-/-^ mice (lacking macrophage LepR) and LepR^wt^;LDL-R^-/-^ (with intact macrophage LepR), suggesting that macrophage-specific leptin receptors do not play a direct role in development of atherosclerotic lesions. Overall, findings from this in vivo study underscore a potential role of non-hematopoietic cells such as endothelial cells (ECs) and smooth muscle cells (SMCs) in leptin-induced proatherogenic responses.

## 7. Leptin and Endothelial Cells

Multiple reviews [89,90,91,92] published in the past 15 years have discussed numerous clinical and animal studies highlighting hyperleptinemia as an important trigger for endothelial dysfunction. Leptin-induced effects on EC (summarized in Figure 3) are mediated via activation of JAK2, IRS2, PI3-K/Akt and MAPK pathways as well as increased nuclear translocation of STAT3. Previous studies using human and murine EC cultures have reported that incubation with high leptin stimulates the expression of several proinflammatory cytokines including monocyte chemoattractant protein-1 (MCP-1), CRP, TNF-α and IL-6 [60,93,94]. Elevated leptin concentrations have also been shown to increase the expression of cell adhesion molecules such as intercellular adhesion molecule-1 and vascular cell adhesion molecule-1 (ICAM-1, VCAM-1), transforming growth factor-beta (TGF-β) and vascular endothelial growth factor-receptor (VEGF-R) on EC [93,95]; these effects are mediated via its interaction with the long form of leptin receptor, in turn activating the molecular mechanisms that lead to inflammatory responses. These effects in turn compromise SMC function ultimately resulting in impaired endothelium-dependent vasodilation, prompting hypertension and atherosclerosis. Epidemiological evidence indicates that, as opposed to normal physiology, chronic hyperleptinemia, typical of the diseased state, significantly augments sympathetic vasoconstrictor activity while blunting endothelium-dependent relaxation [63]. Specifically, in healthy individuals, circulating leptin at physiological levels upregulates endothelin-1 (ET-1)-dependent vasoconstrictor activity concomitant to enhanced nitric oxide (NO)-mediated vasodilation. These counteracting effects of physiological leptin preserves the balance between ET-1 and NO-dependent mechanisms that control vascular homeostasis. In contrast, chronically elevated leptin disrupts the physiological balance between ET-1 and NO-mediated pathways in the vasculature, suggesting a state of leptin resistance in the endothelium. Blockade of such EC-dependent vasodilation in response to hyperleptinemia has been confirmed in coronary vessels and coronary arterioles isolated from lean and obese rodent and canine models [64,96]. Interestingly, Jamroz-Wisniewska et al. [96] have reported that NO deficiency observed in hyperleptinemic obese rats could be somewhat compensated via upregulation of an endothelial-derived hyperpolarizing factor (EDHF), in part mediated via hydrogen sulfide. Notably, in EC cultures isolated from wild-type and eNOS^-/-^ mice, leptin infusion was found to upregulate neuronal NOS (nNOS) expression [97], possibly as a compensatory mechanism to NO deficiency, to maintain endothelium-dependent relaxation. Putative mechanisms underlying such EC-specific leptin resistance may include downregulation of leptin receptor and downstream signaling pathways as well as alterations in receptor binding sites on the endothelium. Additionally, upregulation of caveolin-1 (Cav-1) followed by Cav-1-dependent negative feedback on leptin signaling has been suggested to modulate endothelial leptin resistance under hyperleptinemic conditions [98]. A growing body of data further demonstrates that leptin may trigger oxidative stress via increased generation of reactive oxygen species (ROS) [60,94,95], a primary mediator of endothelial dysfunction. Such an increase in the oxidant stress burden impedes endothelium-dependent relaxation triggering EC dysfunction and atherosclerosis. Some suggested mechanisms for this response include upregulation of NF-kB-dependent proinflammatory and chemotactic pathways, lipid peroxidation and activation of vasoactive mediators. Leptin has also been reported to stimulate VEGF-R2 and cyclooxygenase-2 (COX-2) expression, resulting in augmented EC proliferation and capillary tube formation attributing to its pro-angiogenic function [99]. Chronically elevated leptin in obese subjects has been associated with augmented circulating levels of plasminogen activator inhibitor-1 (PAI-1) [68]. Incubation of coronary EC cultures with leptin in vitro have revealed elevated tissue factor (TF) transcripts and upregulation of PAI-1, at both transcriptional and translational levels [100,101]. These effects have been linked to leptin’s pro-coagulant and prothrombotic properties. More recently, it was shown that EC-specific deletion of LepR resulted in augmented ET-1 expression and endothelial activation contributing to increased neointimal hyperplasia and cellularity [102]. In contrast, obese EC-specific-LepR wild-type mice (with intact leptin signaling) showed reduced neointimal formation. These results were similar to that observed in diet-induced mouse models of obesity, depicting augmented neointimal formation. Further, transcriptional upregulation of ET-1 was found to be mediated via increased DNA binding activity of activator protein-1 (AP-1) to ET-1 promoter [102]. These data provide additional support to the previously conceived notion that hyperleptinemia causes EC-specific signaling defects, triggering endothelial leptin resistance in obesity. Interestingly, lack of SMC-specific LepR had no effect on neointimal growth in this study [102]. While these data would suggest a contribution of EC-specific leptin signaling in development of vascular lesions driven by vascular injury (clinically analogous to in-stent restenosis), future investigations are needed to evaluate the contribution of EC-specific leptin signaling in pathogenesis of atherosclerosis.

## 8. Leptin and Vascular Smooth Muscle Cells

In normal physiology, the primary function of vascular smooth muscle cells (VSMCs) is to regulate vascular tone and provide structural integrity to the vessel wall to enable adequate blood flow to different organs in the body. In a healthy blood vessel, VSMCs typically reside within the medial layer of the vessel wall and are maintained in a quiescent contractile state. However, under pathological conditions following exposure to various proatherogenic stimuli (e.g., diabetes, obesity), VSMCs undergo extensive phenotypic transformation from their ‘healthy’ contractile state to ‘diseased’ synthetic phenotype, with enhanced migratory, secretory and hyperproliferative properties [103,104]. SMC migration and proliferation are key events central to the progression of atherosclerosis. A growing body of literature [39,42,105,106,107,108] supports the growth-promoting effects of leptin on VSMC that may likely attribute to leptin-induced neointimal growth, vessel remodeling and atherosclerosis. Previous studies have reported that leptin stimulates VSMC proliferation via increased cell cycle progression to S and G2/M phases accompanied with augmented ERK1/2 and NF-kB activation [105]. Leptin-induced cell cycle regulation has been shown to be mediated via increased expression of Cyclin D1 and β-catenin (regulator of cell–cell adhesion) concomitant to reduced p21^WAF1/Cip1^ (promotes cell cycle arrest) [107,108] expression. Leptin has also been documented to increase VSMC migration mediated via enhanced matrix metalloproteinase-9 (MMP-9) expression [107]. In a later study using coronary arteries and coronary VSMC isolated from domestic swine, acute and chronic leptin exposures, at obese concentrations, were reported to increase SMC proliferative and contractile responses via Rho kinase-dependent pathway [106]. Global proteomics profiling further revealed that leptin administration markedly altered the expression of multiple proteins associated with calcium signaling (e.g., calreticulin, cAMP-dependent protein kinase type II, tropomyosin) as well as cell growth and proliferation (e.g., myotrophin, myoferlin, fibrillin-1) [106]. These data have suggested an involvement of leptin as an upstream mediator of hypercontractile and proliferative responses in coronary SMC. A recent study [109] further demonstrated that leptin downregulates cofilin-1 expression, an enzyme known to catalyze remodeling of filamentous to globular actin. These results are indicative of a regulatory role of leptin in actin polymerization that may facilitate cell mobility and proliferation. Leptin has also been reported to increase the expression of proinflammatory cytokines such as TNF-α and IL-1β in SMC cultures [109] and increase ROS formation [42]. In accordance with earlier findings, we previously demonstrated that leptin at concentrations relevant to obesity upregulates a potent proatherogenic protein, thrombospondin-1 (TSP-1) expression, in human and murine aortic SMC primary cultures [61]. TSP-1 is an extracellular matrix glycoprotein known to facilitate VSMC migration and proliferation [110]. Data from our laboratory as well as others highlight the role of TSP-1 in vascular biology, with counterbalancing effects in lesion pathogenesis [111]. Specifically, using human aortic SMC primary cultures we demonstrated that incubation with anti-TSP-1 blocking antibody significantly inhibited leptin-induced migration and proliferation compared with cells incubated with leptin alone [61]. Consistent with these in vitro data, exogenous leptin administration failed to increase the expression of proliferating cell nuclear antigen (PCNA, proliferation marker) in the aortic vasculature of TSP1^-/-^ mice vs. wild-type animals [61]. Collectively, these data support a role of TSP-1 in leptin-induced VSMC migration and proliferation. In this same work, we have also shown that TSP-1 upregulation in response to leptin occurs at the level of transcription; specifically, JAK2, ERK and JNK-dependent signaling mechanisms were found to be responsible for leptin-stimulated TSP-1 expression [61]. Cogent to these findings, in a recent publication we further provided direct evidence for a TSP-1-dependent mechanism in leptin-stimulated atherogenesis. Specifically, we found that while exogenous leptin administration (mimicking obese concentrations) led to robust lipid-filled lesions and increased plaque area in ApoE^-/-^ mice, both plaque area and lipid burden were significantly attenuated in leptin-treated TSP-1^-/-^/ApoE^-/-^ mice [46]. We further demonstrated that lack of TSP-1 blocked leptin-induced inflammatory and proliferative SMC lesion abundance in ApoE^-/-^ mice. In addition, while increased PCNA and vimentin (SMC synthetic marker) expression were accompanied with reduced smooth muscle-myosin heavy chain (SM-MHC, SMC contractile marker) expression in leptin-treated ApoE^-/-^, global TSP-1 deletion significantly attenuated PCNA and vimentin together with augmented SM-MHC and calponin (SMC contractile marker) expression; these results were also validated in isolated murine aortic SMC primary cultures incubated with elevated leptin concentrations in vitro. In sum, these findings strongly support a regulatory role of TSP-1 in leptin-induced SMC de-differentiation, which may be responsible for atherosclerotic complications associated with hyperleptinemia.

Clinical studies highlight an association between elevated circulating leptin and impaired arterial distensibility in healthy adolescents [112], independent of other confounding factors such as blood pressure, fat mass and insulin resistance. Elevated serum leptin levels have also been linked to arterial stiffness in CAD and hemodialysis patients [113,114]. Studies conducted in SMC-specific LepR knockout mice have suggested that leptin receptor-mediated pathways modulate impaired vascular relaxation in response to hyperleptinemia [115]. Animal studies using vascular injury mouse models have shown that elevated leptin may promote vascular remodeling and arterial stiffness associated with obesity. Specifically, in vivo and in vitro studies using SMC primary cultures isolated from rodent models have reported an upregulation of multiple extracellular matrix proteins (ECM) including MMP-2, MMP-9, Collagen I, Collagen IV and fibronectin in response to leptin [116,117]. Leptin-induced increase in ECM abundance and activity was shown to be mediated via activation of oxidative stress and PI3K/AKT pathways. Moreover, leptin stimulated tissue fibrosis depicted via increased generation of profibrotic factors such as TGF-β and connective tissue growth factor (CTGF); enhanced superoxide anion production was further suggested to mediate this profibrotic response of leptin [116]. Together, these data reveal an involvement of ECM protein rearrangement in hyperleptinemia-induced vascular fibrosis. The ECM composition is an important determinant of arterial stiffness [118]. Disrupted balance between ECM synthesis and ECM degradation may trigger vascular remodeling resulting in structural, functional and mechanical changes in the vasculature. Matrix metalloproteinases (MMPs) produced by VSMCs are known to degrade ECM proteins such as collagen and gelatin, thereby playing a role in plaque instability [119]. To this end, previous studies have reported plaque-destabilizing properties of leptin, with an increase in locally synthesized leptin detected in SMCs of vulnerable plaques [120]. Specifically, in a mouse model of carotid artery ligation [121], leptin administration increased neointimal hyperplasia accompanied with increased MMP-9 expression; these results were also validated in isolated SMC cultures in vitro. It was further shown that leptin-induced upregulation of MMP-9 occurred via activation of ERK and JNK-dependent mechanisms coupled with increased DNA binding activity of AP-1 to MMP-9 promoter [121]. While these data clearly underline a contribution of leptin to lesion instability in vascular injury models of neointimal hyperplasia, the involvement of leptin in the rupture of advanced atherosclerotic plaques warrants future investigation.

Elevated plasma leptin levels in Type 2 diabetic and obese patients directly correlate with the degree of coronary artery calcification [28]. Animal studies and in vitro experimentation using isolated SMC cultures have confirmed that leptin at pathological concentrations promotes VSMC transformation into bone-forming cells coupled with increased mineralization. Specifically, in Western diet-fed ApoE^-/-^ mice, daily leptin administration for 8 weeks significantly enhanced vascular calcification; this was further accompanied with augmented expression of osteoblastic markers (osteopontin, osteocalcin) [122]. Congruently, in primary bovine aortic SMC cultures, leptin was reported to increase osteopontin and bone sialoprotein expression accompanied with reduced matrix gla protein levels, indicative of augmented osteoblast differentiation. Additional work has shown that the leptin-induced osteogenic effect is mediated via inhibition of GSK-3β signaling [123]. Overall, a collection of in vivo and in vitro results (summarized in Figure 4), including recent data from our laboratory, emphasize a regulatory role of leptin in VSMC activation and phenotypic transformation, central to the pathogenesis of atherosclerosis.

## 9. Conclusions

In summary, a balance between physiological and pathological concentrations of leptin determines its beneficial vs. deleterious effects on the vasculature. Although leptin at physiological levels may exert therapeutic effects, pathophysiological amounts of leptin, mimicking diabetes or obesity, exhibit potent proatherogenic properties. As such, an integrated cell-specific leptin response prompting increased cross-talk between EC, SMC and macrophages mediates the deleterious effects of hyperleptinemia on the vessel wall (summarized in Figure 5). A major strength of this review is the inclusion of clinical studies involving diverse patient cohorts (CAVD, MI, SLE, Type 2 diabetes) presenting elevated circulating leptin levels as well as animal studies ranging from global to cell-specific transgenic murine models. However, we did not discuss the role of perivascular adipose tissue (PVAT)-derived leptin on the pathogenesis of atherosclerotic complications associated with diabetes and obesity, and this constitutes a limitation of this review. While the existing literature indicates contribution of different vascular cell types (SMC, EC, macrophages) to leptin-induced atherogenesis, more systematic approaches using cell-specific LepR knockouts and lineage tracing strategies in the background of atherosclerotic models (ApoE^-/-^, LDL-R^-/-^) are required to elucidate the fundamental molecular mechanisms by which hyperleptinemia drives lesion pathogenesis in obesity and related metabolic complications.

## Figures and Tables

**Figure 1 ijms-22-05446-f001:**
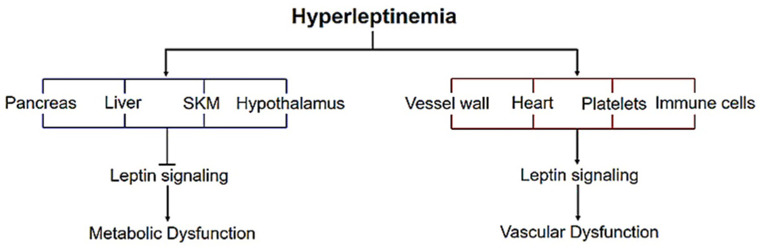
Metabolic vs. vascular dysfunction in response to hyperleptinemia. Elevated leptin levels, representative of obesity and diabetes, inhibit leptin signaling in the pancreas, liver, skeletal muscle (SKM) and hypothalamus, triggering metabolic dysfunction. On the other hand, activation of leptin receptor signaling pathways in the vessel wall, heart, blood platelets and immune cells mediate leptin-induced vascular dysfunction.

**Figure 2 ijms-22-05446-f002:**
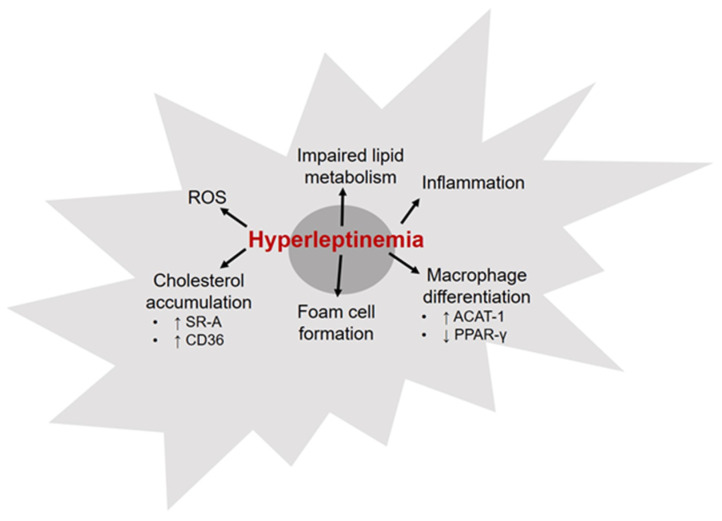
Hyperleptinemia-induced changes on vascular macrophages that contribute to atherosclerosis. ROS: reactive oxygen species; PPAR-γ: peroxisome proliferator-activated receptor-γ; SR-A: scavenger receptor-A; ACAT-1: acyl CoA:cholesterol acyltransferase-1. Please see text for additional details.

**Figure 3 ijms-22-05446-f003:**
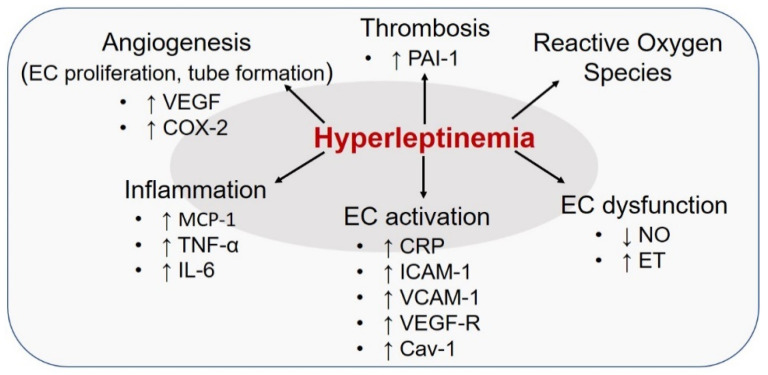
Effect of hyperleptinemia on endothelial cells in the vessel wall triggering atherosclerosis. VEGF: vascular endothelial growth factor; COX-2: cyclooxygenase-2; ICAM-1: intercellular adhesion molecule-1; VCAM-1: vascular cell adhesion molecule-1; MCP-1: monocyte chemoattractant protein-1; TNF-α: tumor necrosis factor-α; IL-6: interleuikin-6; CRP: C-reactive protein; PAI-1: plasminogen activator inhibitor-1; Cav-1: caveolin-1; ET: endothelin-1; NO: nitric oxide. Please see text for additional details.

**Figure 4 ijms-22-05446-f004:**
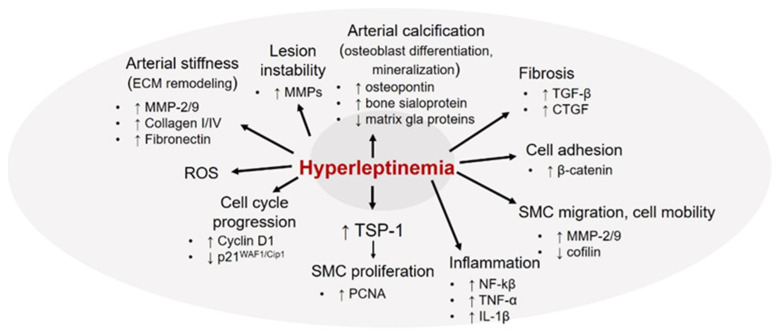
Hyperleptinemia-induced SMC activation and phenotypic changes triggering atherosclerosis. MMP: matrix metalloproteinase; TGF-β: transforming growth factor-β; ECM: extracellular matrix; TSP-1: thrombospondin-1; ROS: reactive oxygen species; CTGF: connective tissue growth factor. Please see text for additional details.

**Figure 5 ijms-22-05446-f005:**
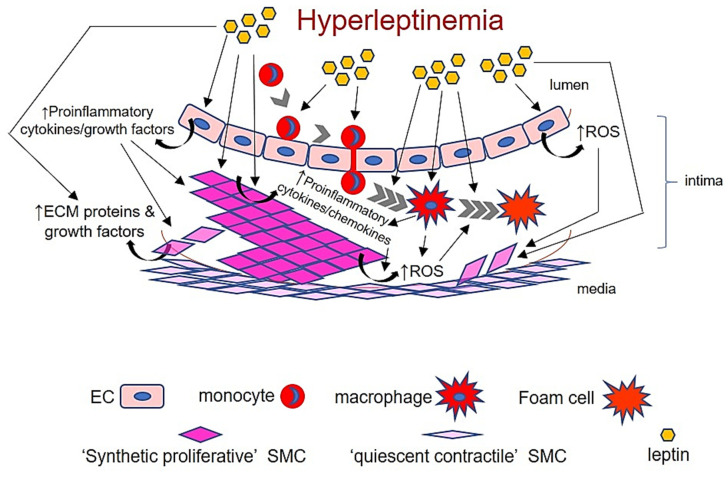
Cross-talk between EC, SMC and macrophages in response to hyperleptinemia in the vessel wall. ECM: extracellular matrix; ROS: reactive oxygen species. Please see text for details.

**Table 1 ijms-22-05446-t001:** Overview of clinical studies defining the role of leptin in vascular dysfunction. CAD: coronary artery disease; CAC: coronary artery calcification; SLE: systemic lupus erythematosus; WC: waist circumference; BMI: body mass index; CRP: C-reactive protein; TC: total cholesterol; TG: total triglyceride; CAVD: calcified aortic valve disease; eGFR: estimated glomerular filtration rate; AMI: acute myocardial infarction; LVEF: left ventricular ejection fraction; CK: creatine kinase; IMT: intima-medial thickness; Gensini score: provides an assessment of the severity of CAD.

Study Population	Major Findings	References
CAD patients (± high fasting serum leptin	↑ leptin levels ∞ ↑ cardiovascular events	[24]
Advanced congestive heart failure Pediatric patients with congenital heart defects undergoing cardiopulmonary bypass surgery	Elevated plasma leptin levels and soluble leptin receptor Cardiopulmonary Bypass Surgery associated with circulating leptin	[25]
Type 2 diabetics in varying plasma leptin quartiles	Higher leptin quartiles ∞ ↑ CAC scores	[28]
SLE patients vs. healthy controls	↑ plasma leptin ∞ ↑ carotid plaque ↑ proinflammatory lipids	[29]
CAD patients vs. healthy controls	↑ leptin; ↑ WC, BMI ↑ leptin ∞ ↑ CRP; ↑ TG, TC ↑ cardiac enzymes	[30]
CAD vs. healthy controls	↑ serum leptin ∞ ↑ Gensini score leptin levels ∞ number of involved coronary vessels	[31]
CAVD vs. non-CAVD patients	↑ serum leptin in CAVD leptin level ∞ age leptin level ∞ 1/eGFR	[32]
AMI patients vs. stable cardiac patients	↑ serum leptin during course of AMI mean serum leptin ∞ LVEF mean serum leptin ∞ CK serum leptin ∞ diseased coronary vessels	[33]
Hypertensive patients (± MI)	serum leptin ∞ MI	[34]
CVD patients vs. controls (meta-analyses including 13 studies)	high leptin levels not associated with risk of CHD or stroke	[35]
Young normotensive healthy adults	circulating leptin ∞ 1/carotid IMT circulating leptin ∞ 1/ (carotid cross sectional wall area)	[36]

**Table 2 ijms-22-05446-t002:** Overview of animal studies on the role of hyperleptinemia in neointimal hyperplasia and atherosclerosis. HFD: high-fat diet; I/M: intima/media; TC: total cholesterol; TSP-1: thrombospondin-1; BCA: brachiocephalic artery.

Animal model	Diet/Treatment	Major Findings	References
FeCl_3_-induced carotid artery injury in WT, ob/ob & db/db	recombinant murine leptin (0.6 µg/g, 3 weeks) ± HFD	↑ neointimal formation in HFD-fed WT vs. ob/ob ↑ injury-induced neointimal thickness and luminal narrowing in leptin-treated WT and ob/ob ↓ neointimal formation in db/db No effect on neointimal growth in leptin-treated db/db	[37]
femoral artery wire injury- induced WT, ob/ob and db/db	recombinant murine leptin (± 0.4 mg/Kg, 2 weeks)	↑ injury-induced neointimal hyperplasia in leptin-treated WT and ob/ob vs. db/db	[38]
injury-induced WT, ob/ob	normal chow; HFD	↑ serum leptin, ↑ neointimal area and ↑ luminal stenosis in HFD-fed WT No neointimal growth in HFD-fed injury-induced ob/ob	[39]
femoral artery wire injury in db/db and WT littermates		↓intimal area; ↓I/M in db/db vs. WT	[40]
femoral artery-induced vascular injury in WT, ob/ob, db/db	recombinant murine leptin (5 µg/g bw) adeno virus-expressing murine leptin	↓neointimal area, ↓I/M ratio in ob/ob and db/db vs. WT following vascular injury ↑neointimal area in leptin-treated or adeno leptin-infected ob/ob vs. untreated ob/ob No effect on neointimal growth in adeno leptin-infected db/db	[41]
FeCl_3_-induced carotid artery injury in ApoE^-/-^, LDL-R^-/-^ and WT	recombinant murine leptin (0.6 µg/g bw)	↑neointimal formation in leptin-treated LDL-R^-/-^ ↑neointima, luminal stenosis in leptin-treated WT No effect on neointimal growth in leptin-treated ApoE^-/-^	[42]
ApoE^-/-^	recombinant murine leptin (125 µg 1x daily for 4 weeks); western diet	↑atherosclerotic lesions in carotid and brachiocephalic arteries No difference in atherosclerotic lesions in ascending aorta	[43]
ob/ob;ApoE^-/-^ vs. ApoE^-/-^	atherogenic diet (16 weeks)	↓atherosclerotic lesions in ob/ob;ApoE^-/-^ vs. ApoE^-/-^ ↑atherosclerotic lesions in leptin-treated ApoE^-/-^	[44]
ob/ob;LDL-R^-/-^ vs. LDL-R^-/-^	± HFD (12 weeks)	↓ aortic sinus lesions in ob/ob;LDL-R^-/-^ vs. LDL-R^-/-^ ↑TG, glucose, insulin levels in ob/ob;LDL-R^-/-^ vs. LDL-R^-/-^	[45]
TSP1^-/-^;ApoE^-/-^ vs. ApoE^-/-^	recombinant murine leptin (125 µg, 1x daily for 3 weeks), ± western diet	↑ atherosclerotic lesions in leptin-treated ApoE^-/-^ vs. untreated ApoE^-/-^ ↓ lipid burden, lesion size in leptin-treated TSP1^-/-^;ApoE^-/-^ vs. leptin-treated ApoE^-/-^	[46]
db/db;ApoE^-/-^, db/+;ApoE^-/-^ and ApoE^-/-^		↑ lesions in db/db;ApoE^-/-^ vs. ApoE^-/-^	[47]
ob/ob;LDL-R^-/-^	recombinant murine leptin (0.1–3 mg/Kg bw, 12 weeks)	↓ aortic root & BCA plaque area, ↓lipid burden in leptin-treated ob/ob;LDL-R^-/-^ vs. untreated ob/ob;LDL-R^-/-^	[48]
Ins2^+/Akita^;ApoE^-/-^ vs. Ins2^+/+^;ApoE^-/-^	recombinant murine leptin (0.4 µg/g bw, 1x daily for 12 weeks)	↓ fasting plasma leptin, ↑ atherosclerotic lesions in Ins2^+/Akita^;ApoE^-/-^ vs. Ins2^+/+^;ApoE^-/-^ ↓ lesion area, ↓TC in leptin-treated Ins2^+/Akita^; ApoE^-/-^ vs. untreated Ins2^+/Akita^;ApoE^-/-^	[49]

## Data Availability

Not applicable.

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
