# Peer review of "Leptin in Atherosclerosis: Focus on Macrophages, Endothelial and Smooth Muscle Cells"

_ijms, 2021, doi:10.3390/ijms22115446_

Round 1

Reviewer 1 Report

The manuscript entitled: “Leptin in Atherosclerosis: Focus on Macrophages, Endothelial

and Smooth Muscle Cells” is an interesting review article that summarized the up to know knowledge related to leptin and its implication in atherosclerosis and cardiovascular diseases. The manuscript is well written, but some issues should be added before publication.

The references should be arranged in the text according to the instruction for the journal.

In the introduction, you have to add a paragraph regarding the aim of this review article.

In the part 3. Leptin-friends or foe in atherosclerosis and vascular dysfunction at 3.1 and 3.2 I suggest adding two tables one that includes all the clinical studies identified that associate leptin levels and atherosclerosis or vascular dysfunctions and the main findings and another for the animal studies.

The strength and limitations of this article should be included.

Reviewer 2 Report

Raman and Khanal provide here an extensive overview of the impact of hyperleptinemia on the development of atherosclerosis. After a brief introduction, authors develop a specific part on clinical studies and animal studies. Then, they describe the effect of leptin signaling in the context of metabolic and vascular dysfunction. This is probably the most interesting part as they hypothesize, based on a thorough review of the current literature, that the hyperleptinemia will preferentially affect the vasculature but not metabolic organs because of their leptin resistance. This is particularly elegant. Then authors review how leptins control the cellular activity of the main cell type involved in atherogenesis, namely endothelial cells, macrophages and SMCs.

Overall, the current report is well-organized, the different sections are well-balanced, the review is very well-written and thoughtful. Current draws are appropriate and provide a resume of the dedicated sections. This review is timely and opens new research pathways that worth publication with the following minor modifications/addings.

We would however recommend to add two tables, one for the clinical studies section and one for the animal studies section. Indeed, authors highlight the existence of discrepancies in the literature. It would be useful for the readership to have these different discrepancies in a table resuming the different studies.

In addition, in the clinical studies section, lane 88, authors should highlight the difference between the different studies in term of cohort composition. Are they comparable to other studies? What may explain the discrepancies? Are patients overweight/obese in all studies? How does the leptin levels evolve compared to obesity in these studies?

Finally, an integrated scheme harboring the deleterious effect of leptin on atherogenesis in the 3 cell types (EC, macrophages, SMC) and emphasizing the different cellular cross-talk would be useful as a concluding drawing.

Round 2

Reviewer 1 Report

The authors addressed all the comments. Now the manuscript is ready for acceptance in the current form. 
